# *Where Am I and What Will I See*: AN AUTO-REGRESSIVE MODEL FOR SPATIAL LOCALIZATION AND VIEW PREDICTION

**Junyi Chen**[1,2]    **Di Huang**[2]*    **Weicai Ye**[2]    **Wanli Ouyang**[2]    **Tong He**[2]

[1]Shanghai Jiao Tong University [2]Shanghai AI Lab

junyichen@sjtu.edu.cn    dihuanginfo@gmail.com

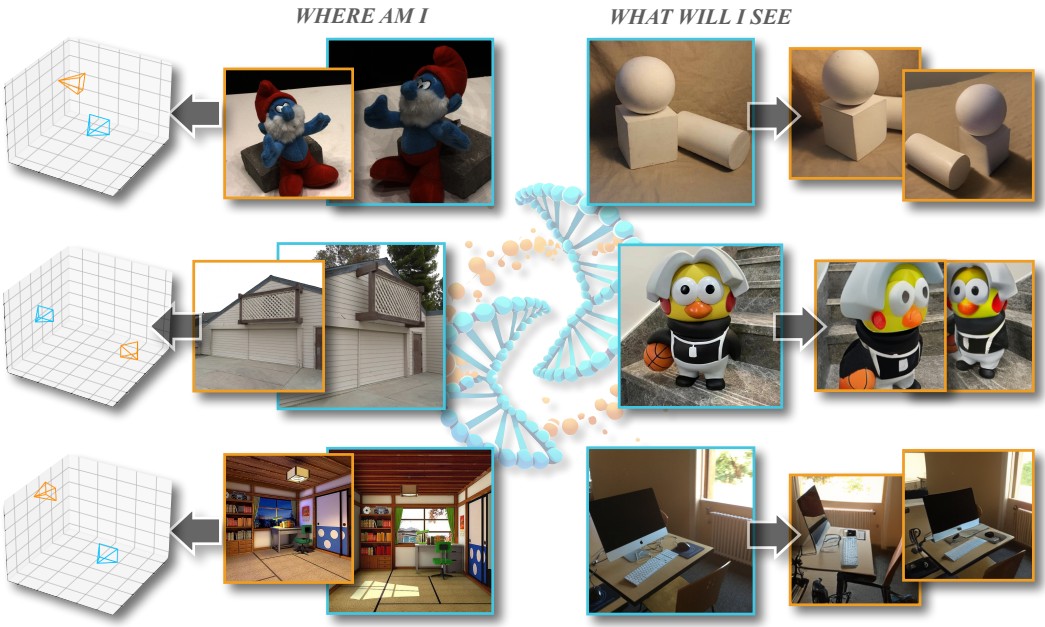

Figure 1: We introduce **G**enerative **S**patial **T**ransformer (GST), which has unified the image representation and camera pose representation within the realm of 3D vision, enabling the autoregressive generation of results in another modality given an observed image and a specific modality.

## ABSTRACT

*Spatial intelligence* is the ability of a machine to perceive, reason, and act in three dimensions within space and time. Recent advancements in large-scale autoregressive models have demonstrated remarkable capabilities across various reasoning tasks. However, these models often struggle with fundamental aspects of spatial reasoning, particularly in answering questions like "Where am I?" and "What will I see?". While some attempts have been done, existing approaches typically treat them as separate tasks, failing to capture their interconnected nature. In this paper, we present **G**enerative **S**patial **T**ransformer (GST), a novel auto-regressive framework that jointly addresses spatial localization and view prediction. Our model simultaneously estimates the camera pose from a single image and predicts the view from a new camera pose, effectively bridging the gap between spatial awareness and visual prediction. The proposed innovative camera tokenization method enables the model to learn the joint distribution of 2D projections and their corresponding spatial perspectives in an auto-regressive manner. This unified training paradigm demonstrates that joint optimization of pose esti-

---

*Corresponding Author.

mation and novel view synthesis leads to improved performance in both tasks, for the first time, highlighting the inherent relationship between spatial awareness and visual prediction. Project page: https://sotamak1r.github.io/gst/.

# 1 INTRODUCTION

Spatial Intelligence is a critical piece of the AI puzzle, as it encompasses the ability to understand the spatial relationships between objects and scenes. "Where am I?" and "What will I see?" are two fundamental questions used to test the spatial capability of intelligent agents. For humans, commencing from the observation of an image, they can infer spatial layouts and predict unseen aspects of the environment. This spatial ability enables humans to easily orient themselves in space and to envision observations from different perspectives within a given space. Therefore, expanding existing general intelligence into the realm of 3D space, enabling it to effectively answer the aforementioned two foundational questions, is a crucial step in the development of spatial intelligence.

Modern auto-regressive models (Brown, 2020; Touvron et al., 2023) demonstrate exceptional intelligence due to their advanced architecture, enabling effective long-range dependency modeling. This capacity empowers large language models (LLM) to exhibit outstanding intellectual performance across various domains (Kondratyuk et al., 2023; Team, 2024). To enable the model to answer "Where am I?" and "What will I see?" effectively, we endeavor to leverage the strong modeling capabilities of auto-regressive models to spatial intelligence. These two questions correspond to two classic tasks in the 3D domain: spatial localization and view prediction. Prior research has traditionally treated the tasks of generating novel views from a given location (Liu et al., 2023; Tewari et al., 2023; Chan et al., 2023) and estimating camera poses from varied perspectives (Wang et al., 2023; Zhang et al., 2024) as distinct tasks, typically employing separate models for each task. Nevertheless, they are closely interconnected in human spatial cognition, as individuals often subconsciously integrate them during spatial reasoning without subjectively distinguishing between them.

To bridge this gap, we propose **G**enerative **S**patial **T**ransformer (GST), a model inspired by the observations of human spatial reasoning processes. Rays entering the eye form image signals; hence, 2D image is the projection of 3D space from a given viewpoint position and direction. Building upon this notion, we introduce, for the first time, the concept of utilizing the camera as a new modality into the training of an auto-regressive model. Specifically, we leverage Plücker coordinates to transform the camera into a camera map akin to an image, and convert it into a token sequence by applying a tokenization method similar to that used for image. To address the uncertainties associated with scene scale and unseen regions, we employ an auto-regressive approach to construct a joint distribution of novel views and camera locations given an initial observation. This joint distribution inherently encapsulates two posterior probability distributions, one for novel views and another for camera locations, and introduces two conditional probability distributions. This approach contrasts with directly modeling two completely different distributions using the same model.

Experiments demonstrate that modeling a single joint distribution leads to a more stable optimization process compared to modeling two distinct distributions. This stability is achieved without compromising the final convergence accuracy, ultimately leading to better results in both novel view synthesis and relative camera pose estimation tasks through the introduction of redundant objectives. Furthermore, by integrating additional target distributions, our model can effectively complete novel tasks such as sampling valid camera poses from an observation, generating images under no camera conditions. This approach has been shown to significantly improve the model's understanding of the intricate nuances present within 3D spatial environments.

In summary, our contribution lies in introducing GST, the first model capable of concurrently performing both novel view synthesis and relative camera pose estimation within a unified framework. Drawing inspiration from human spatial reasoning, we design GST to model the joint distribution of images and camera poses, enabling it to effectively integrate the training objectives of both tasks. Extensive experiments demonstrate that GST achieves state-of-the-art performance in synthesizing a single novel view in a feed-forward manner while accurately estimating the relative camera pose between two frames, establishing a new benchmark for spatial intelligence in vision-based systems.

## 2 RELATED WORK

### 2.1 AUTO-REGRESSIVE MODELS

The core concept of auto-regressive models is to establish the probability distribution of future sequences given the current sequence. Building upon this principle, powerful large language models (Brown, 2020; Touvron et al., 2023) have emerged, demonstrating their ability to tackle a variety of challenging language tasks. To extend these capability to multimodal tasks, researchers have explored tokenization methods (Esser et al., 2021; Yu et al., 2024; Zeghidour et al., 2021) that are specifically tailored to different data modalities. By tokenizing data from various modalities, researchers can construct diverse task sequences and leverage the next-token prediction mechanism of language models for training (Team, 2024; Kondratyuk et al., 2023). This approach has shown promising results in enhancing multimodal capabilities.

### 2.2 NOVEL VIEW SYNTHESIS

In recent years, significant strides (Rombach et al., 2022; Ramesh et al., 2022; Saharia et al., 2022) have been made in the field of image generation. Numerous scholarly investigations posit that these modalities effectively encapsulate intricate 3D prior knowledge. One notable approach, Zero-1-to-3 (Liu et al., 2023), introduces a methodology that begins with a single image and incorporates the relative camera pose as a contextual factor to define the conditional distribution of novel views, yielding promising results. However, generalizing this approach to real 3D environments remains an area requiring further scholarly inquiry. Additionally, DFM (Tewari et al., 2023) conceptualizes the synthesis of new perspectives within a scene as a solution to stochastic inverse problems. Nevertheless, the exorbitant computational demands inherent in the denoising process and the resource-intensive nature of volume rendering impede the seamless scalability of this framework. In contrast, CAT3D (Gao et al., 2024) enhances this process by concatenating the representation of each view's camera using Plücker ray notation with the image channels. It employs a diffusion model to characterize the conditional distribution of multiple views based on a specific camera configuration.

### 2.3 CAMERA POSE ESTIMATION

Estimating camera poses from sparse views poses a significant challenge, as sparse images often lack sufficient information to accurately determine the position and orientation of the camera. COLMAP (Schönberger & Frahm, 2016) ccomplishes camera pose estimation by detecting and matching features in images, estimating relative camera poses using robust algorithms, refining poses through bundle adjustment. This is a computationally intensive process and performs poorly with sparse viewpoints. RelPose (Zhang et al., 2022) addresses this issue by leveraging an energy-based model to amalgamate relative rotations into a set of camera poses. PoseDiffusion (Wang et al., 2023) utilizes diffusion models to directly sample camera parameters. Building upon this foundation, Ray Diffusion (Zhang et al., 2024) represents a notable advancement. While it still employs a diffusion model, this approach generates camera rays as targets, demonstrating superior precision compared to directly predicting camera parameters.

## 3 METHOD

We consider the challenge of simultaneously sampling novel view images and their corresponding camera poses given a single input image. Diverging from prior research, our focus lies in uncovering the inherent consistency between these two tasks rather than alternately training the two objectives during the training process. Our approach starts by tokenizing the image and camera spatial positions, merging two codebooks to ensure the model treats both modalities equally (Sec 3.1). We then proceed to train a generative network to model the joint distribution of these components (Sec 3.2).

### 3.1 TOKENIZATION

**Image Tokenization**. In our approach, we employ VQGAN (Esser et al., 2021) as our image tokenizer, comprising the following components: an encoder that maps an image $x \in \mathbb{R}^{H \times W \times 3}$ to a feature map $f \in \mathbb{R}^{h \times w \times d}$, and a quantizer containing a codebook of $k$ $d$-dimensional vectors. The

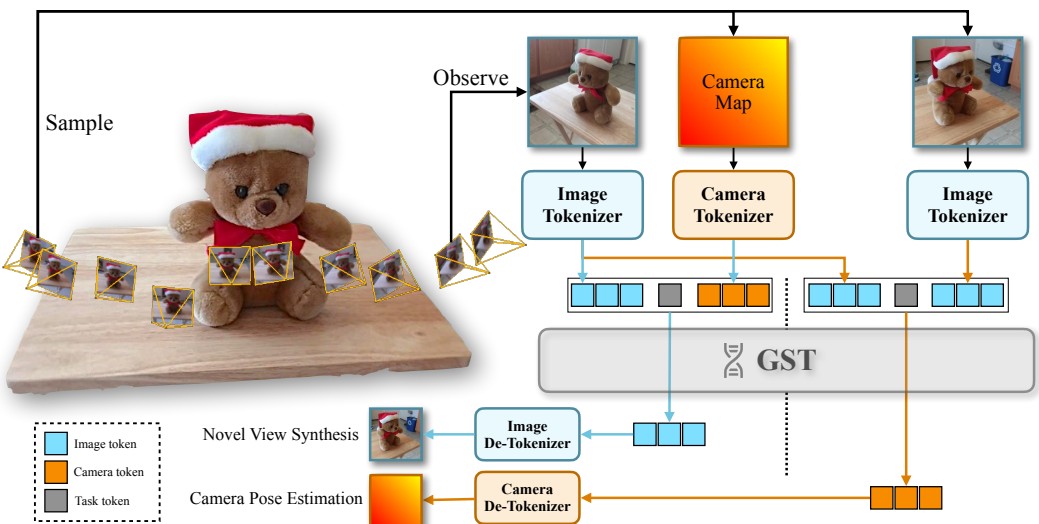

Figure 2: **Illustration of the GST** . Upon providing an observed image, task category, and the target camera position or the other view image, GST autonomously generates the desired outcome in an auto-regressive manner. The training process of GST comprises two significant phases: (1) training of the image and camera tokenizer, (2) training of the auto-regressive model.

quantization process involves selecting the nearest vector from the codebook, based on Euclidean distance, for each $d$-dimensional vector at every position in $f$. The index in the codebook serves as the image encoding, and the quantized feature map $z$ is then reprojected back to the image space $\hat{x}$ through a decoder.

Concerning constraints on the quality of image reconstruction, both $\mathcal{L}_2$ reconstruction loss and LPIPS perceptual loss $\mathcal{L}_p$ are utilized, along with incorporating an adversarial loss $\mathcal{L}_d$. Consequently, a discriminator is alternately trained during this process, with $\lambda_d$ representing the weight of the adversarial loss:

$$\mathcal{L}_x = \mathcal{L}_2(x, \hat{x}) + \mathcal{L}_p(x, \hat{x}) + \lambda_d \mathcal{L}_d(\hat{x}). \tag{1}$$

Due to the non-differentiability of quantization, the straight-through estimator is employed to propagate gradients from the decoder to the encoder:

$$z = \text{sg}[z - f] + f, \tag{2}$$

where $\text{sg}[\cdot]$ stands for the stopgradient operator (Van Den Oord et al., 2017). The training loss of the codebook is defined as the proximity of the embedding of the force codebook to the features output by the encoder. The utilization of the stopgradient operator prevents the gradient from propagating back to the encoder. To address this, a loss term is introduced to enforce the feature vectors extracted from the encoder to approach those in the codebook, with the weighting adjusted by the parameter $\beta$:

$$\mathcal{L}_{vq} = \left\|\text{sg}[f] - z\right\|_2^2 + \beta\left\|f - \text{sg}[z]\right\|_2^2. \tag{3}$$

The final loss function comprises a combination of image reconstruction loss and codebook parameter constraint loss.

**Camera Parameterization**. Liu et al. (2023) directly utilizes the azimuth and elevation changes in camera poses as model inputs, yet such a simplistic parameterization proves challenging to extend to real-world scenarios. Following Zhang et al. (2024), we opt to utilize Plücker rays for the dense parameterization of the camera pose.

Specifically, for a camera extrinsic matrix, the spatial position and orientation of the camera can be derived. Each pixel on the image plane corresponds to a ray emanating from the camera origin,

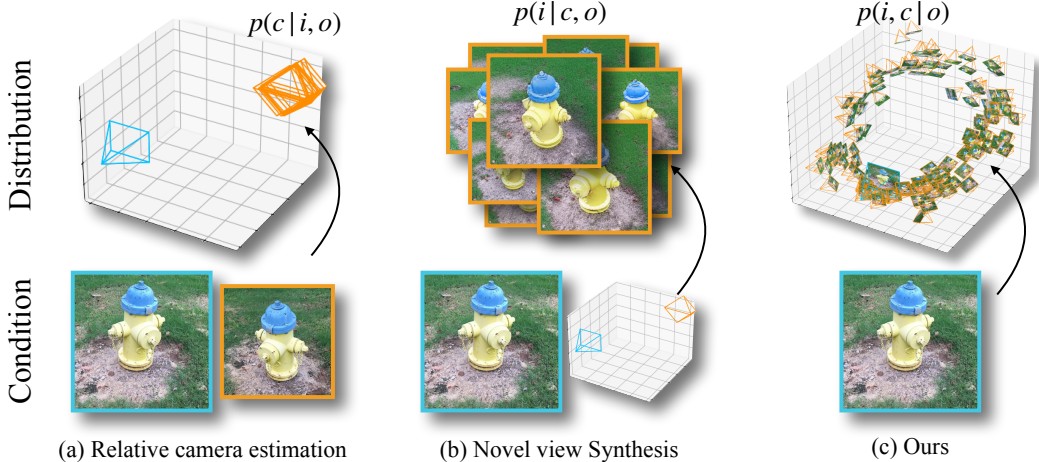

(a) Relative camera estimation     (b) Novel view Synthesis     (c) Ours

Figure 3: **Comparison of target distributions for different tasks**. Previous methods have traditionally constructed unimodal target distributions for novel view synthesis and camera estimation tasks. However, GST has introduced a joint distribution for both the image and the corresponding camera poses.

denoted as $r = (o, d)$. Plücker rays represent this ray as a 6-dimensional set of Plücker coordinates $r = (o \times d, d)$, encapsulating both the position and direction of the ray in space. Upon converting all rays emanating from the camera origin and intersecting with image pixels into Plücker coordinates, we can obtain a tensor of the same size as the image, referred to as the camera map. The camera map can be utilized to infer the camera matrix through a reverse derivation process (Zhang et al., 2024).

**Camera Tokenization**. To enable a auto-regressive model to handle both camera and image modalities concurrently, we have adopted a consistent tokenization method for processing camera rays, akin to image tokenization. This approach involves leveraging a modified version of the VQVAE (Van Den Oord et al., 2017) with reduced convolutional network depths in both the encoder and decoder, following a training procedure analogous to that of the image tokenizer, albeit without the discriminator component. By adjusting the initial size of the camera map, we ensure the eventual derivation of camera tokens that align in size with the image tokens, facilitating seamless integration within the model architecture.

## 3.2 JOINT DISTRIBUTION MODELING

**Backbone**. Previously, we formulated the entire training approach as an auto-regressive problem, which could be addressed using modern large language model (LLM) techniques. Here, we employed the training paradigm of next-token prediction to effectively integrate the capabilities of advanced large language models for future applications. We adopt Sun et al. (2024) codebase, and implemented techniques from current LLM, such as QK-Norm (Henry et al., 2020), RMSNorm (Zhang & Sennrich, 2019), and leveraged the 2D form of rotary positional embeddings (RoPE) (Su et al., 2024) to operate on the image and camera map embeddings.

**Training Target**. In the pursuit of unifying two divergent training tasks $p(i|c, o)$ and $p(c|i, o)$, a straightforward strategy entails the direct alternation between the respective training objectives throughout the training regimen. Nonetheless, the discernible dissimilarity between the disparate probability distributions can precipitate pronounced instability in training dynamics. To redress this quandary, our focus has pivoted towards the joint distribution governing novel view images and camera poses, as illustrated in the figure 3. This distribution inherently encapsulates the training requirements for these dual tasks:

$$p(i, c|o) = p(i|c, o)p(c|o) = p(c|i, o)p(i|o). \tag{4}$$

Upon completing training, during the inference stage, it proves adequate to stochastically sample from the prescribed prior probabilities $p(i|o)$ or $p(c|o)$ under conditional settings (either through manual intervention or automated model-driven processes) to generate results corresponding to the

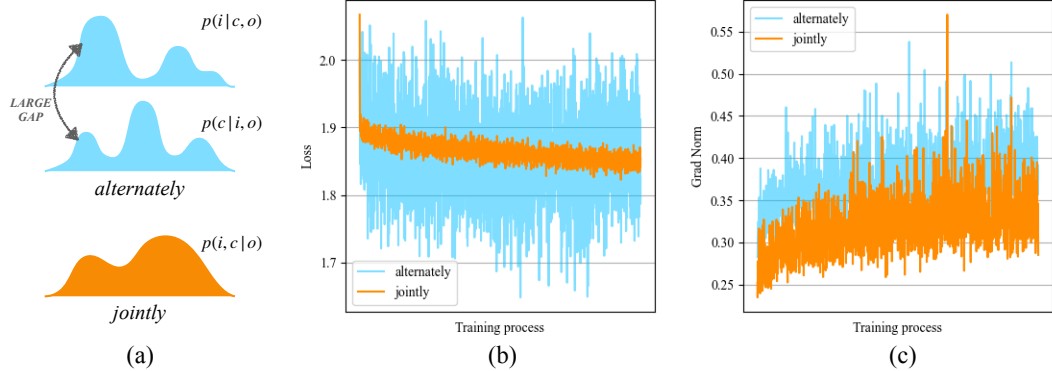

Figure 4: **The impact of joint distribution training**. (a) By transforming two distributions into a joint distribution, the target of the model can be unified. (b) (c) We trained using the same initial model parameters and hyperparameters, averaging the loss and gradient norms every 50 steps, and observed that joint distribution training yields a more stable training trajectory compared to alternatively training two objectives.

alternate modality, thereby realizing the objective of unifying the dual tasks within a singular model. Crucially, the circumvention of the necessity to oscillate between the training of two vastly different training objectives ensures a more stable trajectory in training losses and gradient norms, as illustrated in the figure 4.

In our training stage, we tokenize the initial observed image $o$ along with the image $i$ and camera $c$ corresponding to random sampled viewpoints, resulting in three token sequences ($t_o$, $t_i$, and $t_c$) of equal length. These sequences are concatenated to form a fixed-length token sequence denoted as $s$. Further details will be discussed in the supplementary materials. Subsequently, a decoder-only transformer denoted as $G$ and parameterized by $\theta$ is trained. The generation target is formulated as:

$$\mathcal{L}(\theta) = -\sum_{j=|t_o|+1}^{|s|} log P(s_j|s_1, \ldots, s_{j-1}; \theta) \tag{5}$$

# 4 EXPERIMENTS

## 4.1 EXPERIMENT SETUP

**Datasets**. We trained GST auto-regressive model and camera tokenizer on four datasets with multi-view images and camera pose annotations: Objaverse (Deitke et al., 2023), CO3D (Reizenstein et al., 2021), RealEstate10k (Zhou et al., 2018), and MVImgNet (Yu et al., 2023). These datasets encompass 3D object models, real-world environments, and object-centric scenes. Objaverse exclusively offers object 3D models, for which we utilized the rendering outputs from Liu et al. (2023) and employed the filtered object ID list provided by Tang et al. (2024). We performed center cropping on all training images to ensure obtaining token sequences of the same length.

**Baseline**. We adopted the feed-forward method Zero-1-to-3 (Liu et al., 2023), and its associated models: Zero-1-to-3 XL, trained on the larger-scale dataset Objaverse-XL (Deitke et al., 2024); Zero-1-to-3 NVS, fine-tuned by ZeroNVS (Sargent et al., 2023) on real-image datasets. Our test set comprises two parts: A randomly selected non-overlapping subset from Objaverse, distinct from the training set; And a subset extracted from CO3D, following the settings of ZeroNVS. We selected various methods (Zhang et al., 2024; 2022; Lin et al., 2023; Wang et al., 2023) for evaluating the camera pose estimation performance of GST, following the experimental setup of raydiffusion (Zhang et al., 2024), conducting quantitative experiments on the relative camera pose estimation of 2 images on CO3D (Reizenstein et al., 2021).

**Implementation details**. We utilized the image tokenizer from LlamaGen (Sun et al., 2024) along with its auto-regressive model with 1.4 billion parameters to initialize our model weights, and made some slight modifications to the architecture like adding QK-Norm (Henry et al., 2020) to the attention operations. Throughout the training process, the weights of the image tokenizer were kept constant, while all parameters of the auto-regressive model were trained. We employed a structure

similar to the image tokenizer to construct our camera tokenizer, albeit with fewer parameters. The parameter quantities for each model is detailed in table 1a. The base learning rate for training the camera tokenizer is set at $10^{-4}$ with a batch size of $128$. The auto-regressive model also starts with a base learning rate of $10^{-4}$, which later decreases to $10^{-5}$ in the later stages of training. The batch size for the auto-regressive model is $192$, with gradient accumulation performed every $8$ steps. Both models utilize the AdamW optimizer with a gradient clipping threshold set at 1.0.

Table 1: We present the selection of the model parameter size and the camera tokenizer codebook size that we used for our all experiments.

(a) **Model Parameters.**

| Models | Parameters |
|---|---|
| Image tokenizer | 77 M |
| Camera tokenizer | 22 M |
| Auto-regressive model | 1.4 B |

(b) **Camera Tokenizer Codebook.**

| Size | Dim | usage↑ |
|---|---|---|
| 1024 | 4 | 65.1% |
| 2048 | 2 | 34.6% |
| 2048 | 4 | 90.4% |
| 2048 | 8 | 13.1% |
| 4096 | 2 | 21.3% |
| 4096 | 4 | 77.0% |

## 4.2 NOVEL VIEW SYNTHESIS

We conducted a comparison between GST and the comparative methods (Liu et al., 2023; Sargent et al., 2023) on Objaverse and CO3D. As shown in the figure 5 and table 2, even though GST was trained on a subset of Objaverse, it achieved superior results compared to Zero-1-to-3, which was trained on a several orders of magnitude larger dataset than ours.

This advantage is also evident in the quantitative metrics displayed in the table 2.

## 4.3 RELATIVE CAMERA POSE ESTIMATION

We tested GST on the CO3D (Reizenstein et al., 2021) benchmark to evaluate its performance in estimating relative camera poses. Here, we trained the model using a setting with only two frames, as this setting is considered the most challenging yet fundamental for this task. As shown in the table 3, we achieved the best generalization under this setting (Unseen Categories). Given the joint training of GST across multiple datasets, a subsampled CO3D training set was utilized to optimize training efficiency. This may explain the slightly lag behind in the accuracy of training categories compared to the approach presented in Zhang et al. (2024).

## 4.4 ABLATION STUDY

**Camera Tokenizer Designs**. The size of the camera tokenizer's codebook and the length of the quantized vectors are crucial factors that influence the performance of subsequent models. We trained our camera tokenizer on the same dataset used for training the auto-regressive model, instead of randomly sampling camera positions in space. While the latter approach could better represent all camera positions in space, it would reduce the usage of the camera tokenizer's codebook during the training of the auto-regressive model. This is because some randomly sampled, unconventional camera positions would consume a portion of the tokenizer's training resources, even though these positions would not appear during auto-regressive model training. Moreover, we aim for the model

Table 2: **The quantitative results of novel view synthesis**. GST outperforms Zero-1-to-3, Zero-1-to-3 XL in terms of LPIPS and SSIM metrics on Objaverse dataset. However, Zero-1-to-3 achieves higher PSNR scores compared to GST due to its training on the complete Objaverse dataset, which includes our test set. Furthermore, the GST significantly outperforms Zero-1-to-3 NVS across all three quantitative metrics, attributable to our carefully crafted camera condition.

| | LPIPS ↓ | PSNR ↑ | SSIM ↑ |
|---|---|---|---|
| **Objaverse Dataset** | | | |
| Zero-1-to-3 | 0.135 | **14.77** | 0.845 |
| Zero-1-to-3 XL | 0.141 | 14.53 | 0.834 |
| GST (ours) | **0.085** | 13.95 | **0.871** |

| | LPIPS ↓ | PSNR ↑ | SSIM ↑ |
|---|---|---|---|
| **CO3D Dataset** | | | |
| Zero-1-to-3 NVS | 0.515 | 13.4 | 0.407 |
| GST (ours) | **0.419** | **15.6** | **0.456** |

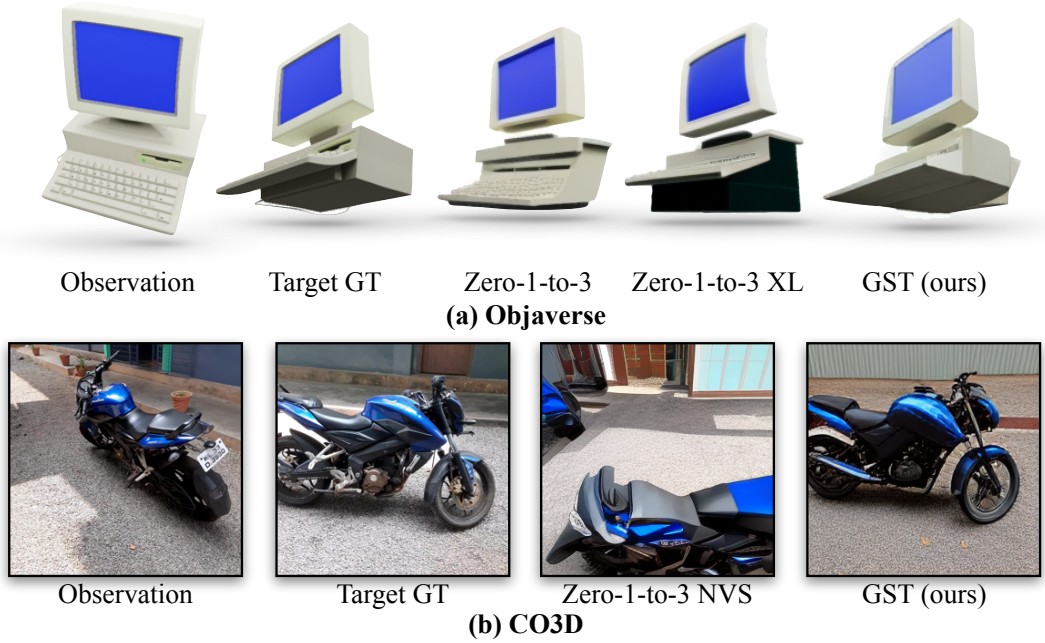

Figure 5: **Visualization comparison of novel view synthesis**. Among these methods, GST demonstrates the highest quality of generated results. Additionally, due to the design of the camera condition, GST is capable of producing the most accurate images for the specific viewpoints.

Table 3: **Camera Rotation Accuracy on CO3D (@ 15°).** Here we report the proportion of relative camera rotations that are within 15 degrees of the ground truth. Results for all comparative methods are referenced from Ray Diffusion.

| Methods | Seen Categories | Unseen Categories |
|---|---|---|
| RelPose | 56.0 | 48.6 |
| PoseDiffusion | 75.7 | 63.2 |
| RelPose++ | 81.8 | 69.8 |
| Ray Diffusion | **91.8** | 83.5 |
| GST (Ours) | 86.6 | **85.1** |

to learn a camera distribution that aligns with the dataset's distribution. This alignment is expected to help the model gain an spatial understanding from observations, as mentioned below.

In the table 1b, we present the usage of the codebook under different parameters. As the distribution of cameras can be better learned compared to images, we observed very low reconstruction losses for all parameters. Consequently, usage became a critical selection criterion. It is notable that as the size and dimension of the codebook increase, we observe a trend of increasing usage followed by a decrease, aligning with observations from previous work Sun et al. (2024). Ultimately, we selected parameters with a size of 2048 and a dimension of 4 for our final model.

**Different Camera Parameterizations**. In Section 3.1, we mentioned that we represent the camera using a method based on Plücker coordinates. Prior to this, we had explored another direct tokenization approach: representing the relative camera's rotation matrix as a three-dimensional vector in terms of Euler angles, and adding an offset to ensure all elements of this vector are positive. After quantization into positive integers, we can then use a codebook of size 360 to represent each dimension. Ultimately, only 3 tokens are needed to represent this rotation angle. We only conducted this ablation experiment on a subset of the object-level dataset Objaverse, as we did not incorporate transformations of camera positions for the sake of simplification. As shown in the results depicted in the figure 6, it is evident that using only 3 tokens as a condition in the auto-regressive model for novel view synthesis makes it challenging to generate ideal viewpoints. In other words, there is a need to introduce more detailed conditions to improve the quality of the generated outputs.

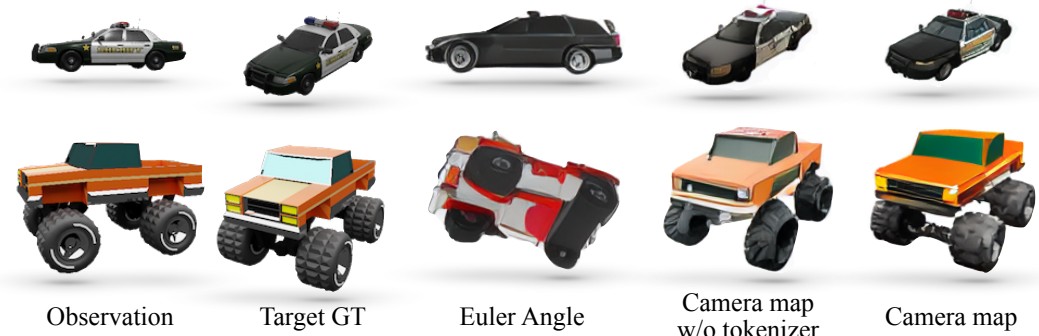

|  |  |  |  |  |
| --- | --- | --- | --- | --- |
| Observation | Target GT | Euler Angle | Camera map w/o tokenizer | Camera map |

Figure 6: Through our experiments with different camera conditions, we observed that employing token-wise conditioning and tokenizing the cameras alongside the images yielded the best generation results for the auto-regressive model.

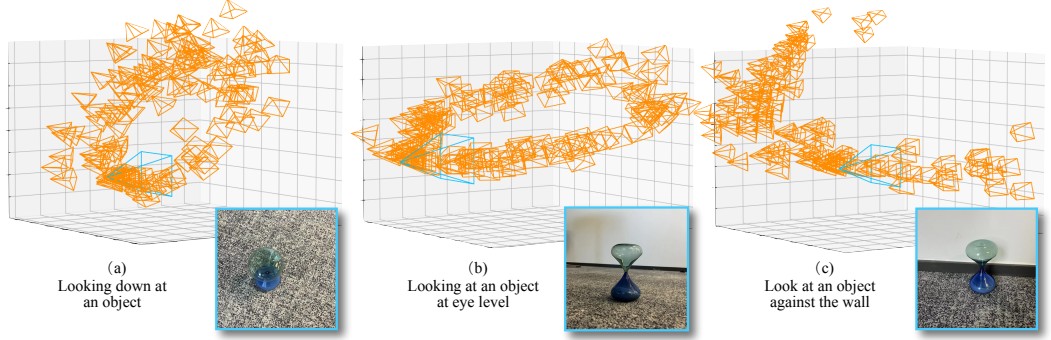

| (a) Looking down at an object | (b) Looking at an object at eye level | (c) Look at an object against the wall |

Figure 7: We present the results of automatically sampled camera distributions using GST, obtained from observing the same object placed in different locations and under various perspectives. The figure showcases three distinct scenarios: (a) a top-down view of the object, where the sampled cameras are predominantly top-down; (b) a frontal view of the object, where the sampled cameras are primarily horizontal; and (c) the object positioned near a wall, where the sampled camera locations effectively avoid the obstruction.

We also explored a scenario where only the camera is used as a condition without requiring the model to output the camera. Specifically, instead of training a camera tokenizer, the Plücker coordinates obtained from the camera are encoded using sine and cosine functions and fed into a small MLP (Mildenhall et al., 2020). This MLP, initially trained concurrently with the auto-regressive model, was subsequently frozen after a predetermined number of training steps. As shown in the figure 6, we obtained visual results from this model, indicating that tokenization continues to exhibit the best performance in the generative process of the auto-regressive model.

**Joint Distribution Training *vs* Alternately Training**. As mentioned in the section 3.2, training a joint distribution tends to have a more stable training trajectory compared to alternating between training two separate targets. This stability is crucial in the training of large transformers, as instability in loss can lead to loss explosion during auto-regressive training, aligning with our experimental observations. Furthermore, we have quantitatively compared the two approaches in the table 4, revealing that training on the joint distribution outperforms alternately training for the selected two

Table 4: we compared the performance of two training methods on two distinct tasks. Our findings indicate that training the joint distribution leads to superior results compared to alternating training on the two distributions.

|  | *Pose Estimation* |  | *Visual Prediction* |  |  |
| --- | --- | --- | --- | --- | --- |
|  | @ 15° ↑ | @ 30° ↑ | LPIPS ↓ | PSNR ↑ | SSIM ↑ |
| GST (alternately) | 84.6 | 92.8 | 0.554 | 10.97 | 0.328 |
| GST (jointly) | **85.7** | **93.5** | **0.500** | **11.90** | **0.358** |

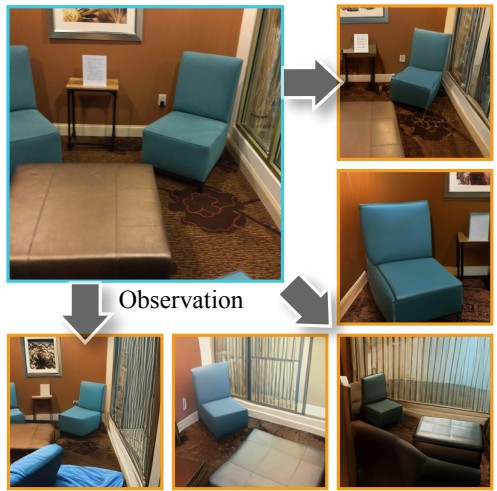

Figure 8: **Uncondition image generation**. Training unconditional image generation in each scenario can assist the model in constructing the distribution of images within that context.

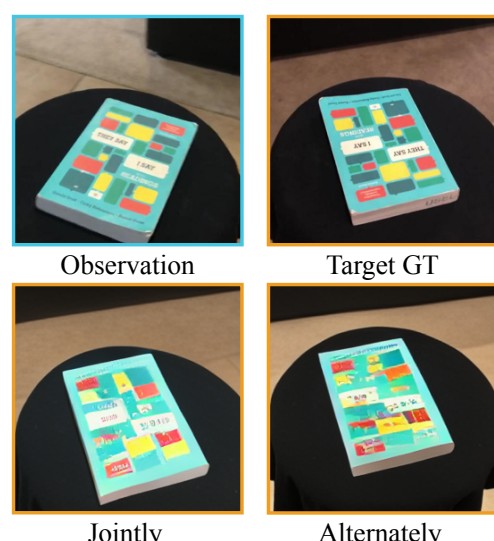

Figure 9: By employing training on the joint distribution, GST exhibits enhancements in image quality and spatial positional accuracy.

tasks. This observation underscores the notion that establishing a well-defined objective distribution can positively impact both the training process and the ultimate outcomes.

**Introduction of Excess Distribution**. As mentioned in Section 3.2, despite introducing an additional distribution, our model not only avoids performance degradation but also achieves enhanced performance while ensuring training stability. A straightforward interpretation is that our conditional prior distribution further assists the model in truly understanding the space.

As illustrated in the figure 7, we capture a real-world object from various angles and positions, allowing GST to sample valid camera distributions from $p(c|o)$ for each scenario. For images captured from a top-down perspective (a), GST predominantly sampled cameras with a top-down viewpoint. Similarly, for objects viewed from a frontal angle (b), GST preferentially sampled cameras with a frontal perspective. Notably, in scenarios involving obstacles (c), GST effectively avoided these obstructions and sampled reasonable camera positions. These results, achieved without any manual intervention, further demonstrate GST's ability to accurately comprehend the spatial layout from observed images.

As depicted in the figure 8, we employ an alternative distribution for sampling $p(i|o)$ by GST, which compels the model to learn the unconditional image distribution within the same scenario. This approach, as discussed in Ho & Salimans (2022), intertwines unconditional loss within the conditional generation training process to strike a balance between sample quality and diversity. This conclusion has also been validated in the figure 9.

## 5 CONCLUSION

In this paper, we propose the Generative Spatial Transformer (GST). To the best of our knowledge, this is the first work that connects novel view synthesis with camera pose estimation. Our method treats the camera as a bridge between 2D projections and 3D space by introducing a camera tokenizer and including the camera as a new modality in training the auto-regressive model. Furthermore, we propose a joint distribution as the training target, enabling diverse task completion and boosting the model's spatial understanding. These advancements not only broaden the model's capabilities but also set the stage for future strides in spatial intelligence.

ACKNOWLEDGMENTS

This work was done during Junyi Chen's internship at Shanghai Artificial Intelligence Laboratory. This work is supported by the National Key R&D Program of China (2022ZD0160102), and Shanghai Artificial Intelligence Laboratory.

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
