# *Where Am I and What Will I See*: AN AUTO-REGRESSIVE MODEL FOR SPATIAL LOCALIZATION AND VIEW PREDICTION

## A ADDITIONAL EXPERIMENTAL DETAILS

### A.1 PRE-PROCESSING

**Camera System Unification**. In our experiments, due to the utilization of diverse camera systems across different datasets, we initially unify the camera coordinate systems of all datasets into a common reference frame, the RUB coordinate system.

**Standardization of Camera Distribution**. Since the camera positions obtained by COLMAP (Schönberger & Frahm, 2016) lack scale information and the datasets used in training stage encompass both synthetic object datasets and real-world scene datasets, significant variations in camera position scales exist across different datasets and scenes. To mitigate the scale discrepancies among different scenes, we first employ a fixed intrinsic camera matrix. Although this approach may introduce certain perspective issues, it does not impact the final results of camera positions and orientations.

Subsequently, we computed the variance of camera positions across different scenes and scaled the camera positions of scenes within the same dataset by a common factor $\beta$ such that the variance of camera positions in the dataset is standardized to 1. We present the $\beta$ corresponding to different datasets in the table 1. This standardization process compacts the relative camera positions within the training set, facilitating the modeling of the overall camera distribution of the dataset.

Finally, as our cameras are randomly sampled, cases where two distant cameras capture images with little to no overlap are prevalent. To address this, during the training of the camera tokenizer and auto-regressive model, we filter out such instances by setting a distance threshold $\delta = 5$ to restrict excessive distances between two cameras.

Table 1: **Dataset scaling factor.**

| Dataset | Scaling Factor $\beta$ |
|---|---|
| Objaverse | 1.0 |
| CO3Dv2 | 0.1 |
| MVImgNet | 0.5 |
| RealEstate10K | 10.0 |

### A.2 DATASET SELECTION

**Objaverse** (Deitke et al., 2023). We incorporated a multi-view subset of Objaverse rendered by Zero-1-to-3 (Liu et al., 2023) as part of our training set. However, this dataset significantly outweighed the remaining real-world image dataset. To enable the model to better comprehend the spatial distribution of real-world scenes, we opted to utilize only a subset of Objaverse. Upon investigation, we observed substantial variance in the rendering quality of different objects within Zero-1-to-3. Thus, we selected a subset of higher quality renderings from Tang et al. (2024) to enhance the efficiency and efficacy of the training process.

**CO3Dv2** (Reizenstein et al., 2021). We selected a subset of categories from CO3Dv2 as our training categories (seen categories), while another subset was designated as unseen categories. The categorization of classes was guided by Ray Diffusion (Zhang et al., 2024), as shown in the table 2.

**MVImgNet** (Yu et al., 2023) **and RealEstate10K** (Zhou et al., 2018). Both datasets consist of real-world scene data, encompassing object-centric scenes and authentic indoor environments. We integrated the complete data from these two datasets into our training process.

Table 2: **Partition of CO3Dv2** (Reizenstein et al., 2021) .

| Seen Categories | | | | | | Unseen Categories | |
|---|---|---|---|---|---|---|---|
| apple | backpack | banana | baseballbat | baseballglove | bench | ball | book |
| bicycle | bottle | bowl | broccoli | cake | car | couch | frisbee |
| carrot | cellphone | chair | cup | donut | hairdryer | hotdog | kite |
| handbag | hydrant | keyboard | laptop | microwave | motorcycle | remote | sandwich |
| mouse | orange | parkingmeter | pizza | plant | stopsign | skateboard | suitcase |
| teddybear | toaster | toilet | toybus | toyplane | toytrain | | |
| toytruck | tv | umbrella | vase | wineglass | | | |

### A.3 AUTO-REGRESSIVE MODEL TRAINING

**Task Tokens**. In this study, we focus solely on two tasks: novel view synthesis and camera pose estimation. Therefore, our task tokens are limited to these two, placed at the end of the codebook for easier future expansion with additional tasks.

**Training Process**. At the initial phase, we evenly allocate training resources among four conditional distributions. Subsequently, we observe that reducing the occurrences of $p(i|o)$ and $p(c|o)$ during training gradually enhances the numerical outcomes. However, this approach also entails a trade-off by compromising a portion of training stability.

**Attention Mechanisms**. In our training stage, we tokenize the initial observed image $o$ along with the image $i$ and camera $c$ corresponding to random sampled viewpoints, resulting in three token sequences of equal length. These sequences are concatenated to form a fixed-length token sequence denoted as $s$. The concatenation order of the three tokens plays a crucial role in how the model interprets and integrates the information. Two concatenation orders, were employed, as illustrated in Figure 1, denoted as $(t_o, t_i, t_c)$ and $(t_o, t_c, t_i)$.

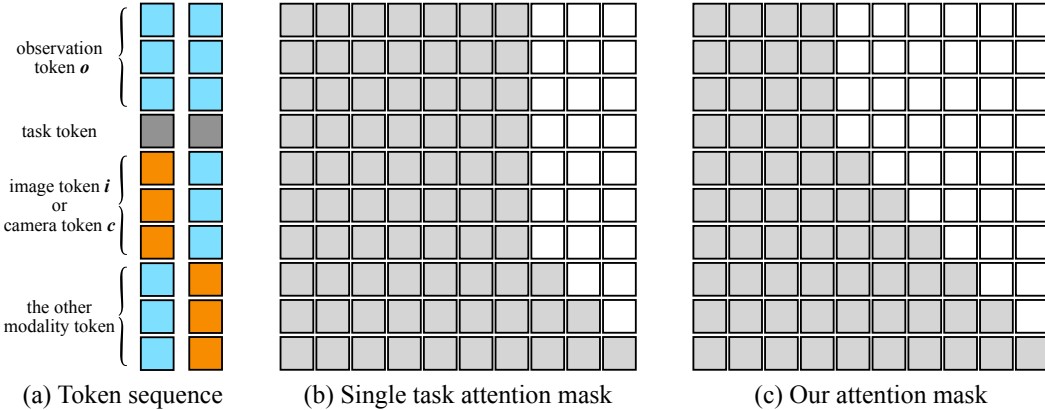

(a) Token sequence     (b) Single task attention mask     (c) Our attention mask

Figure 1: **The comparison of Attention Mechanisms.** (a) We standardize the token sequences $s$ of two tasks to the same length. (b) In the conventional alternating training scheme, target modality tokens can attend to all preceding conditional tokens. (c) We propose to model the joint distribution such that, in (a), the token sequences for the two tasks have visibility only to the currently observation tokens $o$.

The attention mask in figure 1 illustrates the distinction between our training approach and alternating training of two targets.

**Training Resources**. Our camera tokenizer was trained for approximately 2 days on 4 NVIDIA A100 GPUs, while our autoregressive model was trained for about 3 weeks on 16 NVIDIA A100 GPUs.

## B    VISUALIZATION RESULTS

### B.1    NOVEL VIEW SYNTHESIS

We selected a number of representative images, including those from the training dataset, virtually synthesized images, real-world images, and stylistic images, as initial observations. Due to the uncertainty regarding the scale of the scenes, we first employed GST sampling to determine reasonable camera positions. These positions were then used as conditions in conjunction with the initial observations to generate new perspective images, as illustrated in the figure 2.

### B.2    RELATIVE CAMERA POSE ESTIMATION

We selected several highly challenging examples to test the spatial localization capabilities of GST. As illustrated in the figure 3, the selected image pairs include real-world images, images of the same subject taken under different shooting conditions, and images of the same object depicted under various artistic styles. GST demonstrated outstanding performance across all these examples.

## C    LIMITATIONS AND FUTURE WORKS

The scarcity of multi-view datasets with precise camera annotations poses a significant barrier to scaling up GST. In the current work, we only explored the most fundamental scenario involving a single observation image and one novel perspective. Consequently, when sampling multiple images and camera positions simultaneously, issues of consistency may arise, although this problem decreases as the number of training viewpoints increases. In the future, we will aim to collect more multi-view data with precise poses and will explore extending GST to accommodate an arbitrary number of views as conditions, thereby broadening its applicability. Additionally, we trained on datasets without scale, and the potential for extension to scenes with real-world scale remains to be investigated.

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

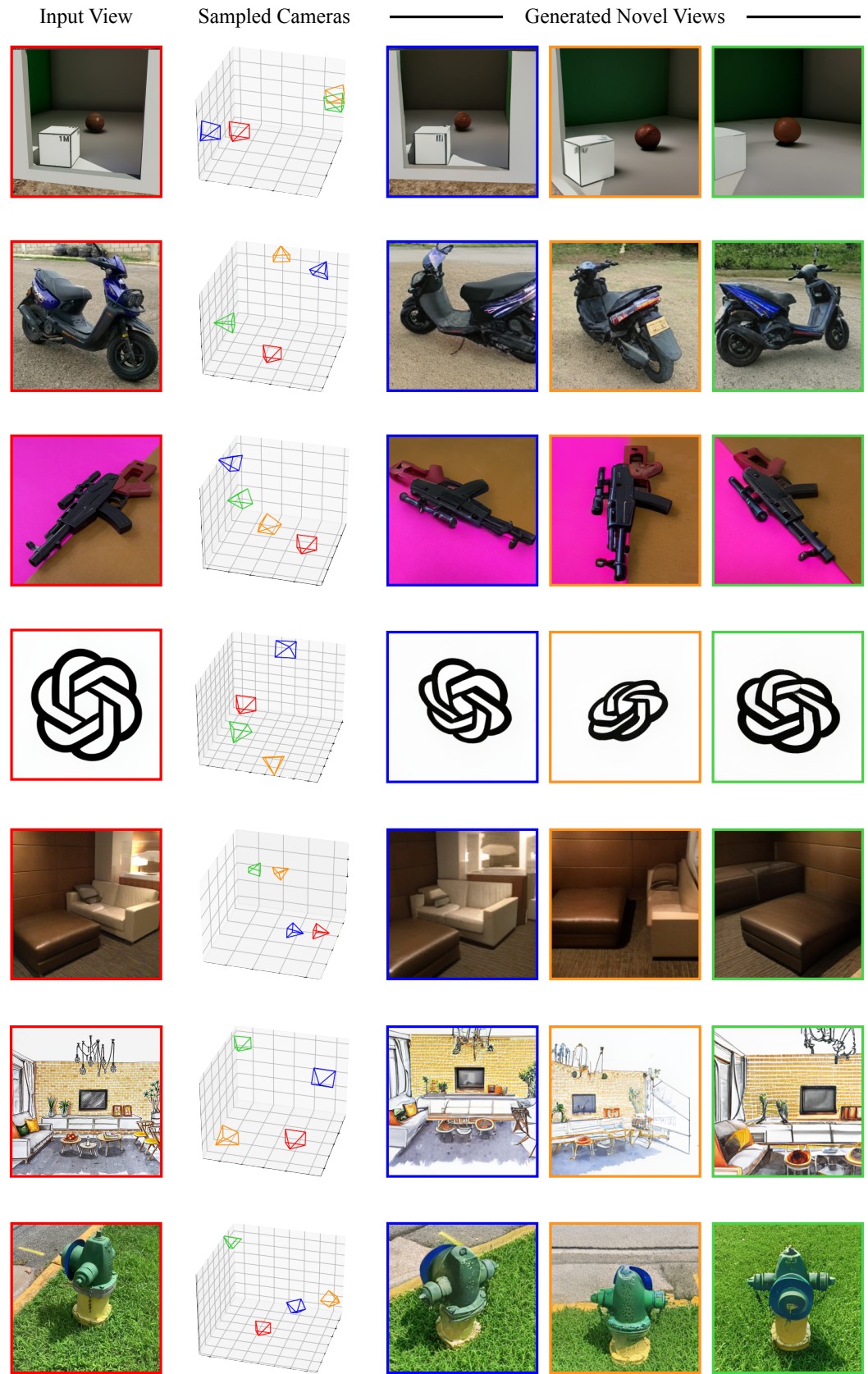

Figure 2: **Visualization results of novel view synthesis**.

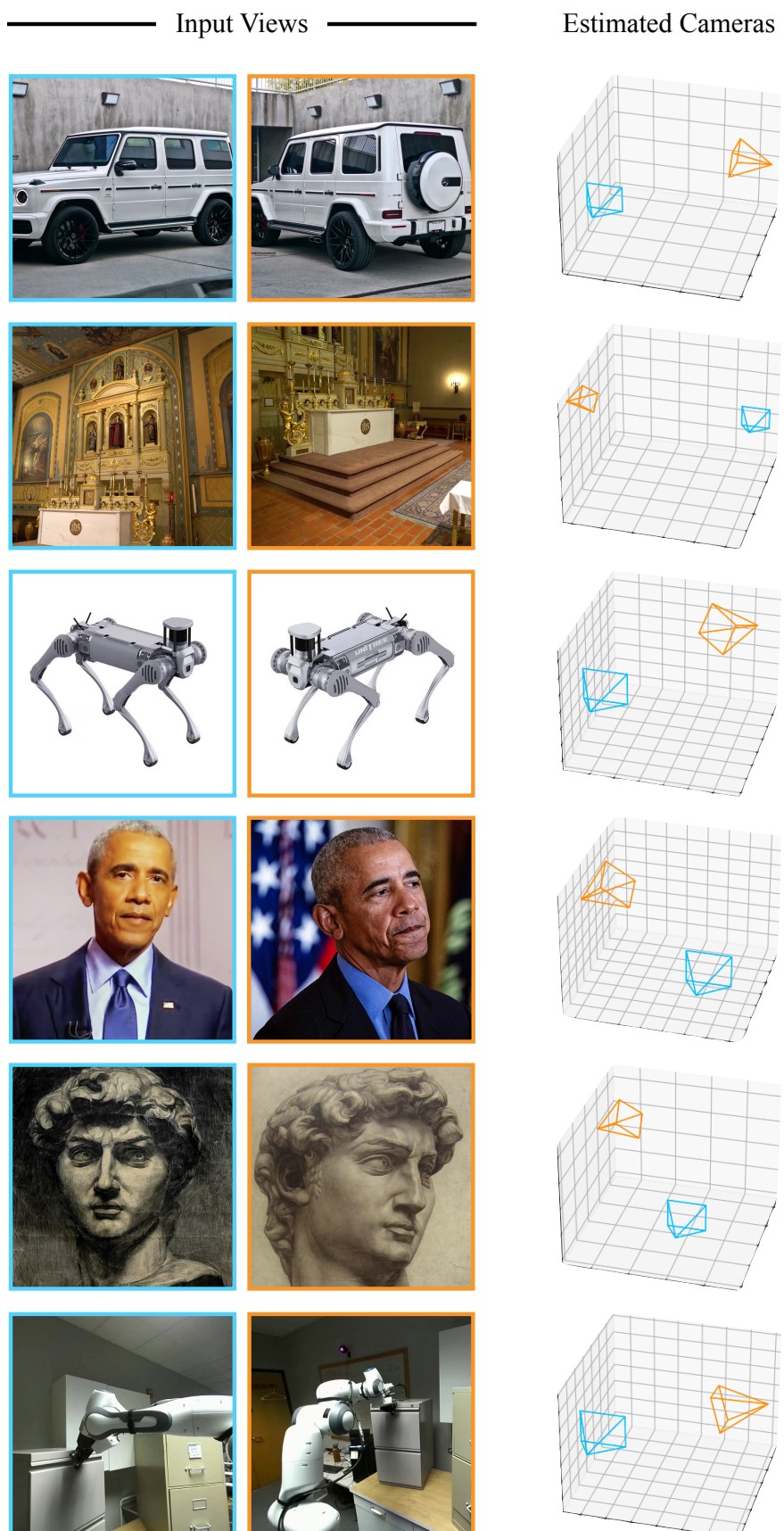

Input Views     Estimated Cameras

Figure 3: **Visualization results of relative camera pose estimation**.