# OpenReview forum: "Where Am I and What Will I See: An Auto-Regressive Model for Spatial Localization and View Prediction"
_ICLR.cc/2025/Conference — ICLR 2025 Poster_

### Official Review · Reviewer_xnP6 · 2024-11-02

**Soundness:** 3
**Presentation:** 3
**Contribution:** 3
**Rating:** 8
**Confidence:** 4

**Summary:**

The authors introduced a auto-regressive framework to simutaneously estimate camera pose of a given image, and synthesize novel views at new camera poses. The authors achieve this via modeling the joint distribution of camera frustums and their image projections in a unified training paradim.

**Strengths:**

1. How the authors leverage VQVAE to tokenize camera views consistent with image tokenization is considered novel.
2. The effective adoptation of language models for autoregressive image/camera pose predictictions is interesting.
3. The choice to model the joint distribution of camera views and frustums within the same model seems clean and effective in combining the task of novel view synthesis and camera estimation.
4. Descent visual illustrations and writing.

**Weaknesses:**

1. The NVS results seems to be mostly on par or slightly better than existing state of the art models only based on the training of "a subset of objaverse," and I am curious of the particular choice of the data subset used in training. What's stopping the authors from training on the same dataset as Zero-1-to-3 to make a more fair comparison?
2. The camera rotation accuracy metric of 15 degree is rather a relatively coarse metric, and the authors did not present the camera translation accuracies in any kind.

**Questions:**

1. As the results are presented as training only on a subset of Objaverse, do the authors have any expectations on how the method will scale?
2. For other camera rotation accuracies (5 degree or 10 degree), will GST's performance still be competitive to state of the art methods?
3. What's the rough GPU memory and latency performance of the proposed pipeline relative to existing work (would this be a potential downsides of autoregressive modeling)?
4. Would it be possible to leverage the model's understanding of the joint distribution of camera angles and views for 3D reconstruction/NVS from a set of potentially uncaliberated images?

---

> ### Author Response · Authors · 2024-11-20
> **Response to reviewer xnP6**
>
> We sincerely thank you for your thoughtful evaluation of our work and for highlighting both the strengths and areas for improvement. You raised some concerns and we have addressed them point by point below:
>
> **1. Dataset Choice and Fair Comparison**
>
> *Response:* Thank you for bringing up this critical issue regarding the fairness of the experimental comparison. However, there are two primary reasons for this setup:
>
> - Firstly, upon inspection, inconsistencies were noted in the rendering quality of objects within Objaverse shared by Zero-1-to-3. Consequently, we selected a subset with relatively higher quality for training.
> - Secondly, our goal was for the model to learn more about the spatial distribution in real-world scenes. Given that the scale of the Objaverse dataset is significantly larger than other real-world scene datasets, we reduced the proportion of Objaverse within the overall training set.
>
> Experimental results indicate that despite Zero-1-to-3 being fine-tuned on the powerful base model[1] and utilizing a larger amount of training data, GST still achieved optimal performance, as illustrated in the table below.
>
> ||Dataset Scale|LPIPS|PSNR|SSIM|
> |:-|:--|:-:|:-:|:-:|
> |Zero-1-to-3| 0.8M|0.135|**14.77**|0.845|
> |Zero-1-to-3 XL|10.2M|0.141|14.53|0.834|
> |GST|0.4M(0.08M is from Objaverse)|**0.085**|13.95|**0.871**|
>
> **2. Camera Rotation Accuracy and Translation Metrics**
>
> *Response:* We agree that providing metrics at finer granularity would offer a more comprehensive evaluation. In response, we have conducted additional experiments to assess the rotation accuracy at 5-degree and 10-degree thresholds.
>
> | |Seen|Categories|Unseen|Categories|
> |:-|:-:|:-:|:-:|:-:|
> | |@5$^\circ$|@10$^\circ$ |@5$^\circ$|@10$^\circ$|
> |Ray Diffusion|25.0| 58.9|23.0|53.8|
> |GST|25.0| **59.9**|**26.4**|**57.0**|
>
> The results indicate that GST even surpasses or maintains competitive performance compared to Ray Diffusion at these tighter thresholds.
>
> To calculate camera translation accuracy on a dataset without a standard scale, it is essential to compute the scene scale for normalization using the center and farthest camera coordinates. This process is meaningless for two frames since, after normalization, the calculated camera translation accuracy would always be 100%, as demonstrated in Table 2 of the Ray Diffusion paper. Therefore, we did not present this metric in our paper.
>
> **3. Expectations on Method Scalability**
>
> *Response:* In fact, our training was not limited to just a subset of Objaverse but encompassed four datasets, as discussed in section 4.1. And the original paper focused on testing the performance of NVS on Objaverse where the scale of the scene and camera positions were definite (as illustrated in appendix A of Zero-1-to-3). Regarding scalability, we observed a significant improvement in results with larger model parameter sizes. Additionally, we believe that incorporating more high-quality datasets with precise camera poses can further enhance results. This aspect will be part of our future work, focusing on scaling up in terms of model parameter size and dataset scale and quality.
>
> **4. GPU Memory and Latency Performance**
>
> *Response:* In our implementation, the GST samples images and corresponding camera positions during inference (batch size=16), requiring approximately 9 GB of GPU memory and 35 seconds. I do not consider this to be a downside of autoregressive modeling: Currently, accelerating autoregressive generation is a hot research topic. Various acceleration methods are being explored, such as generating multiple tokens at once instead of sequentially [2], or utilizing more efficient tokenizers to reduce the number of tokens [3]. In the future, a balance point can be achieved between inference cost and generation quality while harnessing the potent sequential dependency modeling capabilities of autoregressive models.
>
> **5. Leveraging the Model for 3D Reconstruction/NVS from Uncalibrated Images:**
>
> *Response:* GST has the capability to infer camera parameters solely from image data. By aggregating information from multiple uncalibrated images, this functionality can be extended to 3D reconstruction/NVS. We are enthusiastic about this prospect as existing experiments have shown promising results. The current GST is trained only on a two-frame setting, hence there may be inconsistencies in sampling across different batches. In the future, we plan to train GST on multi-view images to enhance consistency between different views, and we will address this in the future work.
>
> We thank the reviewer again for their insightful feedback. We look forward to incorporating these improvements and believe they will demonstrate the full potential of the GST model.
>
>
> [1] Rombach et al: High-resolution image synthesis with latent diffusion models, 2022.
>
> [2] Chang et al. Maskgit: Masked generative image transformer. 2022.
>
> [3] Yu et al. An Image is Worth 32 Tokens for Reconstruction and Generation. 2024.

---

> > ### Comment · Reviewer_xnP6 · 2024-11-25
> >
> > Thanks for your response. The authors have mostly addressed my questions by providing additional experimental results and offering good reasonings regarding the dataset selection, translation metrics, etc. I recommend incorporating this information into the camera-ready version, and I am happy to maintain my rating of 8.

---

> ### Author Response · Authors · 2024-11-26
>
> Dear Reviewer,
>
> Thank you very much for the response! We are grateful to hear that our feedback has addressed your concerns. Once again, we appreciate your review of our paper. We welcome any further discussion!
>
> Best regards,
>
> GST Authors

---

### Official Review · Reviewer_3EtM · 2024-11-03

**Soundness:** 2
**Presentation:** 3
**Contribution:** 3
**Rating:** 6
**Confidence:** 4

**Summary:**

This paper proposes to jointly model the distribution of camera poses and novel views via a shared Generative Spatial Transformer (GST).

**Strengths:**

1. The idea of jointly modeling the distribution of camera poses and novel views via a Generative Spatial Transformer is cool and novel.
2. The writing is clear, and the figures are nice.
3. This paper presents comprehensive ablation studies to justify model designs.

**Weaknesses:**

1. Insufficient evaluation:
    1. Novel view synthesis: The model has been trained on real-world images (CO3D, RealEstate10k, MVImgNet), but the evaluation is only on Objaverse (where the object shows simple and unrealistic textures). Can authors (quantitatively and qualitatively) compare to other baselines in terms of novel view synthesis on CO3D as well? One possible baseline is Zero123, fine-tuned on real-world images by ZeroNVS [1].
    2. Multi-image condition: Can the proposed approach synthesize novel views/estimate camera poses conditioned on multiple images? It'd be interesting to see its flexibility by showing these results (e.g., comparing with baselines, say Ray Diffusion with 3-8 images).
    3. How does the proposed approach compare with the state-of-the-art method, DUSt3R [2]?
2. Unfair comparison: In Tab.3, the proposed method achieves higher accuracy on unseen categories than the baseline. However, this could be explained by the more training datasets used by the proposed method, while the baselines, e.g., Ray Diffusion, are trained only on CO3D. Can the authors train the model on the same data used by Ray Diffusion and compare with it?
3. Minor issue: I think the two arrows in Tab.4 are in the wrong direction (Rot@15, Rot@30).


[1] Sargent et al. ZeroNVS: Zero-Shot 360-Degree View Synthesis from a Single Real Image, 2024.

[2] Wang et al. Dust3r: Geometric 3d vision made easy, 2024.

**Questions:**

1. How does the proposed method deal with scale differences of camera poses from different datasets? Is any normalization used for ground truth camera poses across datasets?
2. Can authors provide more details about training resources? For example, how many GPUs are used, and how long has the model been trained?

---

> ### Author Response · Authors · 2024-11-20
> **Response to reviewer 3EtM**
>
> We sincerely thank the reviewer for their thoughtful and constructive feedback on our manuscript. We are pleased to know that you found our idea of jointly modeling the distribution of camera poses and novel views via a GST both cool and novel. Your appreciation of the clarity of our writing, the quality of our figures, and the comprehensiveness of our ablation studies is greatly encouraging.
> We have carefully considered your comments and suggestions, and provide point-by-point responses below:
>
> **1. Insufficient Evaluation:**
>
> - Novel view synthesis
>
> *Response:*  We agree that including evaluations on real-world datasets would strengthen our work by demonstrating the effectiveness of GST in more realistic settings. In response, we compare our model on CO3Dv2 with Zero-1-to-3, which was fine-tuned by ZeroNVS on real-world images. As shown in the table below, GST outperforms the baseline across three metrics, demonstrating the superior performance of GST in the NVS task.
>
> | |PSNR|SSIM|LPIPS|
> |:-:|:-:|:-:|:-:|
> |ZeroNVS|13.4|0.407|0.515|
> |GST|**15.6**|**0.456**|**0.419**|
>
> These results have been included in the revised manuscript (Tab.2) to provide a comprehensive evaluation of our model.
>
> - Multi-Image Conditioning
>
> *Response:*  Thank you for emphasizing the significance of assessing our model's adaptability in multi-image scenarios. However, being the first attempt to utilize an autoregressive model to model images and camera poses distribution, we focused solely on training the most fundamental setting where a single image serves as the condition. Exploring settings with multiple images as conditions is a direction we plan to investigate in the future. We aim to experiment with more rational model configurations and more efficient training methods to achieve this goal. These considerations have been outlined as part of our future work in the revised paper (Appendix C).
>
> - Comparison with DUSt3R
>
> *Response:* The camera pose estimation using only 2D data struggles to match the performance of DUSt3R, which utilizes 3D data as supervision, and the volume of data employed by DUSt3R significantly exceeds that of GST. Nevertheless, we conducted a quantitative evaluation on the DTU dataset[1], which is not included in the training data for any of the comparative methods. As shown in the table below, GST achieved comparable results, but DUSt3R, leveraging 3D supervision and the optimization-based PnP method for camera position estimation, still attained optimal outcomes.
>
> ||@15$^\circ$|@30$^\circ$|
> |:-|:-:|:-:|
> |Ray Diffusion|75.1|92.0|
> |DUSt3R|97.5|97.7|
> |GST|85.7|93.5|
>
> **2. Unfair Comparison**
>
> *Response:* Thank you for highlighting the importance of fair comparisons. Our aim is to train a universal spatial intelligence on a large-scale dataset, hence we did not train on a small dataset but directly tested the final model. Nevertheless, we try to train our model from scratch on CO3Dv2. This process takes time, and we will update our results during the discussion period.
>
> **3. Minor Issue**
>
> *Response:* Thank you for pointing out the mistake in Table 4. We have corrected the direction of the arrows for Rot@15 and Rot@30 in the revised manuscript.
>
> **4. Handling Scale Differences**
>
> *Response:*  Dealing with scale differences across datasets is indeed a critical aspect when training a model on multiple datasets with varying camera pose conventions. We tackle this issue by implementing a normalization step on the ground truth camera poses during preprocessing. Specifically, we calculate the variance of camera positions across different scenes and scale the camera positions within the same dataset by a common factor $\beta$ to standardize the differences in camera positions within the dataset to a unit variance. In the table below, we present the corresponding $\beta$ values for different datasets.This normalization process compresses the relative camera positions within the training set, facilitating the modeling of the overall camera distribution within the datasets.
>
> |  |Scaling Factor $\beta$ |
> |:-:|:-:|
> |Objaverse|1.0 |
> |Co3D| 0.1|
> | MVImgNet | 0.5 |
> | RealEstate10K | 10.0 |
>
> We have included this crucial preprocessing detail in the revised paper (Appendix Tab.1). Thank you for bringing this to our attention.
>
> **5. Training Resources**
>
> *Response:* Our camera tokenizer was trained for approximately 2 days on 4 NVIDIA A100 GPUs, while our autoregressive model was trained for about 3 weeks on 16 NVIDIA A100 GPUs. These specifics have been added to the revised manuscript (Appendix A.3) to offer transparency regarding the computational resources necessary for reproducing our experiments.
>
> We are grateful for your valuable feedback, which has helped us identify areas for improvement in our work. We believe that incorporating your suggestions will significantly enhance the quality and impact of our paper.
>
> [1] Jensen et al: Large scale multi-view stereopsis evaluation, 2014.

---

> > ### Comment · Reviewer_3EtM · 2024-11-24
> >
> > I appreciate the authors' efforts, and the response addresses most of my concerns. Based on this, I would like to raise my score to 6.
> >
> > That said, I have a few follow-up questions:
> >
> > 1. Could the authors elaborate on the "consistency issue" mentioned in L134 of the appendix? While I understand that presenting results for multi-image conditioning might be impractical at this stage, I’m curious to hear their perspective: Could the current model handle multiple images/cameras as input conditions with minimal modifications (e.g., by simply adding multiple image and camera tokens as inputs)? If not, what are the challenges involved? Regarding training data, I believe DUSt3R has demonstrated scalable data collection methods (e.g., leveraging 8 datasets for training).
> >
> > 2. I would appreciate it if the authors could include the comparison with Ray Diffusion trained solely on CO3D.
> >
> > 3. Did the authors explore the DUSt3R-style normalization approach, where scale normalization is applied to each training sample, coupled with a scale-invariant loss? If not, it might be worth considering its potential impact.

---

> > > ### Author Response · Authors · 2024-11-29
> > > **Response to reviewer 3EtM**
> > >
> > > Thank you for your continued interest in our work. We have carefully considered your questions, and provided a response below:
> > >
> > > **1-1. Consistency Issue**
> > >
> > > We apologize for any unnecessary misunderstandings caused by our phrasing. Our intended meaning was that "*issues of inconsistency may arise.*"
> > >
> > > The GST establishes the conditional distribution $p(i,c|o)$, where $i$ represents the image from a new perspective, $c$ denotes the relative position of that new perspective, and $o$ corresponds to the current observation.
> > > When we simultaneously sample multiple perspectives from this distribution, for a region that is not observed in $o$, the projection under the sampled perspective $c_1$ ​ yields $i_1$, while the projection under the sampled perspective $c_2$ ​ yields $i_2$ . Given that **the sampling of different perspectives is independent**, even though $i_1$ ​and $i_2$ ​ may refer to the same spatial location, their semantics may differ.
> > >
> > > **1-2. Multi-Image Conditioning**
> > >
> > > In order to use multi-view images as conditions and sample new perspectives, the model needs to be able to sample from the distribution $p(i,c|o_1,o_2,c_{1\rightarrow 2},\cdots,o_k,c_{1\rightarrow k})$. However, the existing GST model does not include this distribution because GST generates images and the corresponding camera positions depend only on the observation $o$, without considering an additional viewpoint.
> > >
> > > Therefore, to use multi-view images as conditions, a direct approach is to have GST fit the distribution $p(i_1,c_1,i_2 ​ ,c_2 ​ ,⋯,i_k ​ ,c_k ​ ∣o)$, where $c_m$ ​ represents the relative camera position of the image  $i_m$ ​ with respect to observation $o$.
> > > In this way, the model can encompass a series of distributions: $p(i_1 ​ ,c_1 ​ ∣o)$, $p(i_2 ​ ,c_2 ​ ∣o,i_1 ​ ,c_1 ​)$, $\cdots$, $p(i_k ​ ,c_k ​ ∣o,i_1 ​ ,c_1 ​ , \cdots,i_{k−1} ​ ,c_{k−1} ​ )$, enabling the task of using $1$ to $k$ images as conditions and sampling new perspectives. Specifically, we need to construct the sequence $(o,c_1 ,i_1 ​ ,c_2 ​ ,i_2 ​ ,⋯,c_k ​ ,i_k ​)$ and incorporate it into the training paradigm of GST's next-token prediction. However, if the number of conditioning viewpoints is too high, challenges such as excessively long training times and significant memory requirements may arise.
> > >
> > > **2. Comparison with Ray Diffusion trained solely on CO3D**
> > >
> > > We trained our model solely on the seen categories of the CO3D training set and evaluated the model on the test set, yielding the following results:
> > >
> > > | |Seen Category|Unseen Category|
> > > |:-: |:-: |:-: |
> > > |**Ray Diffusion**|91.8|83.5|
> > > |**GST**|82.0|80.1|
> > >
> > > For the above results, we have summarized the following reasons:
> > >
> > > - We tokenized images into sequences using VQVAE. Specifically, for a $256 \times 256$ image, we represented it with only $16 \times 16 = 256$ tokens. This inevitably leads to **the loss of image texture information**, which is crucial for camera pose estimation based purely on visual input.
> > >
> > > - Before feeding the tokens into the GST, we convert them into embeddings by querying an embedding table. This process may lead to the following issue:
> > > For objects not encountered in the training dataset, if the tokens representing such objects appear rarely or not at all during training, the embeddings corresponding to these tokens will remain in an unoptimized state. Consequently, the embeddings representing different objects may exhibit significant discrepancies. The model thus lacks sufficient information about these tokens, ultimately leading to erroneous outcomes. T**his limitation appears to be a drawback of auto-regressive models: they struggle to generalize beyond training data when the dataset is relatively small.** However, as the amount of data increases, the model can better capture the relationships within the embedding table, leading to improved performance.
> > >
> > > We hope our explanation provides further clarity regarding our model.
> > >
> > > **3. DUSt3R-style Normalization Approach**
> > >
> > > We did not employ the DUSt3R-style normalization approach. Our current strategy involves global normalization of the dataset. Specifically, during data preprocessing, normalization is applied to the camera distribution across the entire scene, rather than individually normalizing each sampled camera pair.
> > >
> > > In the autoregressive model's next-token prediction paradigm, we predict the probability distribution of the next token, not the entire sequence of tokens directly. Since an image or a camera is represented by multiple tokens, we cannot detokenize camera tokens back to 3D space at each step. Consequently, we only utilize cross-entropy loss at the token level as our loss function.
> > > Moreover, our scene-wide camera distribution normalization enables the model to develop a holistic understanding of spatial scale solely from the image, as shown in Figure 7.
> > >
> > > We sincerely appreciate your thoughtful question and hope that our response adequately addresses your concern. We warmly welcome any follow-up discussion!

---

### Official Review · Reviewer_AKff · 2024-11-04

**Soundness:** 3
**Presentation:** 3
**Contribution:** 3
**Rating:** 8
**Confidence:** 4

**Summary:**

This paper introduces the Generative Spatial Transformer (GST), an innovative autoregressive framework capable of simultaneously handling spatial localization and view prediction tasks. By introducing a novel camera tokenization method, GST learns the joint distribution of 2D projections and spatial perspectives during training, thereby improving the performance of camera pose estimation and novel view synthesis. Experiments demonstrate that GST achieves state-of-the-art performance in these tasks, highlighting the intrinsic connection between spatial awareness and visual prediction.

**Strengths:**

First, the paper's approach and the scientific questions it raises are novel and intriguing.
Second, the method proposed in the paper can simultaneously estimate camera pose from a single image and predict the view from a new camera pose, effectively bridging the gap between spatial awareness and visual prediction. Interesting validations are shown in Figure 7.
Finally, the writing of the paper is clear and accessible, with excellent explanations from the motivation to the introduction of the method, and the provided conceptual diagrams are easy to understand.

**Weaknesses:**

I did not find any obvious shortcomings. I only have one suggestion: when citing the methods of other articles, it would be better for the author to briefly introduce them.

**Questions:**

My main concerns are as follows:
Line 268. I am curious about the effect of the concatenation order of the three tokens on the experimental results in this problem.
Line 342, GST only uses part of the data set for training, but I want to know what the results will be if the same experimental settings are maintained as Zero-1-to-3.
In line 375, the author does not elaborate on the definition of Unseen Categories. Of course, I can understand that due to the experimental settings, there may be some disadvantages compared to these methods.

---

> ### Author Response · Authors · 2024-11-20
> **Response to reviewer AKff**
>
> Thank you for your thorough and thoughtful review of our paper. We are delighted that you find our approach novel and intriguing, and we appreciate your positive comments on the clarity of our writing and the effectiveness of our conceptual diagrams. We are grateful for the opportunity to address your suggestions and answer your questions:
>
> **1. Brief Introduction of Cited Methods:** You suggested that when citing methods from other articles, we should provide brief introductions to enhance understanding.
>
> *Response:*  Thank you for this valuable suggestion. We agree that providing brief introductions to the methods we cite will enhance the clarity and accessibility of our paper. In the revised manuscript, we will include concise descriptions of the key methods referenced.
>
> **2. Effect of Concatenation Order (Line 268):** You inquired about how the concatenation order of the three tokens affects the experimental results.
>
> *Response:*  The concatenation order of the three tokens plays a crucial role in how the model interprets and integrates the information. During our training process, two concatenation orders were employed, denoted as  $(t_o, t_i, t_c)$ and $(t_o, t_c, t_i)$.
> Here, $(t_o, t_c, t_i)$ is utilized for visual prediction tasks, while $(t_o, t_i, t_c)$ is employed for spatial localization tasks. During training, these two sequences occur randomly, with the requirement that the GST can output both $(t_i, t_c)$ and $(t_c, t_i)$ sequences given a specific observation $t_o$, using distinct task tokens to specify the token order of the output sequence.
> Consequently, during inference, either condition $(t_o, t_c)$ or $(t_o, t_i)$ can be inputted, leading GST to generate results for the other modality $t_i$ or $t_c$ . Moreover, by solely inputting observations $t_o$, GST automatically samples images $t_i$ and corresponding camera poses $t_c$.
>
> Detailed explanations regarding this aspect have been included in the revised paper (Appendix A.3), aiming to alleviate any uncertainties readers may have on this matter. Thank you for your inquiry.
>
> **3. Comparison with Zero-1-to-3 Under Same Experimental Settings (Line 342):** You wanted to know the results if GST were trained using the same experimental settings as Zero-1-to-3, given that GST only uses part of the dataset for training.
>
> *Response:*  Thank you for bringing up this critical issue regarding the fairness of the experimental comparison.
> However, there are two primary reasons for this setup:
>
> - Firstly, upon inspection, inconsistencies were noted in the rendering quality of objects within Objaverse shared by Zero-1-to-3. Consequently, we selected a subset with relatively higher quality for training to enhance the model's training efficiency.
> - Secondly, our goal was for the model to learn more about the spatial distribution in real-world scenes. Given that the Objaverse dataset is significantly larger in scale compared to the other real-world scene datasets, we reduced the proportion of Objaverse within the overall training set.
>
> Experimental results indicate that despite Zero-1-to-3 being fine-tuned on the powerful base model [1] and utilizing a larger amount of training data, GST still achieved optimal performance, as illustrated in the table below. We attribute this success to the carefully crafted camera condition. Leveraging the scaling capabilities of autoregressive models, if the volume of data were increased, GST would still attain similarly optimal results.
>
> |    | Dataset Scale|LPIPS| PSNR| SSIM|
> |:-|:--|:-:|:-:|:-:|
> |Zero-1-to-3| 0.8M | 0.135 | **14.77** | 0.845 |
> |Zero-1-to-3 XL|10.2M |0.141 | 14.53 | 0.834 |
> |GST| 0.4M (0.08M is from Objaverse) | **0.085** | 13.95 | **0.871** |
>
> **4. Definition of Unseen Categories (Line 375):** You noted that we did not elaborate on the definition of "Unseen Categories" and how this might affect the comparison with other methods due to experimental settings.
>
> *Response:*  We apologize for the lack of a clear definition of "Unseen Categories" in the original manuscript.
> The classification between seen categories and unseen categories is based on the setting provided by Ray Diffusion. We have included this clarification in the revised paper (Appendix Tab.2) to prevent unnecessary confusion.
>
> Once again, we are grateful for your positive evaluation of our work and your constructive feedback. Your insights have helped us identify areas where additional clarification and information will enhance the quality of our paper.
>
> [1] Rombach et al: High-resolution image synthesis with latent diffusion models, 2022.

---

> > ### Comment · Reviewer_AKff · 2024-11-26
> >
> > Thank you for your detailed and thoughtful rebuttal. I greatly appreciate the effort and dedication you have put into addressing the comments and improving the manuscript. Your responses clearly demonstrate a strong commitment to the quality of your work .

---

> > > ### Author Response · Authors · 2024-11-27
> > >
> > > Dear Reviewer,
> > >
> > > Thank you very much for the response! We are grateful to hear that our feedback has addressed your main concerns. Once again, we appreciate your review of our paper. We welcome any further discussion!
> > >
> > > Best regards,
> > >
> > > GST Authors

---

### Official Review · Reviewer_VC2z · 2024-11-05

**Soundness:** 1
**Presentation:** 4
**Contribution:** 2
**Rating:** 3
**Confidence:** 4

**Summary:**

This paper aims to do view prediction and pose estimation with a single autoregressive model. The paper claims that humans do this task effortlessly, and that their own model does it effortlessly too. The method is inspired by LLMs, by which the authors mean: it is autoregressive. The paper claims to be the first to tokenize camera data. The model itself is enormous: >1.4B parameters. It achieves good results.

**Strengths:**

This paper's figures are very well designed. Good color coordination across the paper.

**Weaknesses:**

The paper says that, given an image of a scene, humans can "effortlessly reconstruct the entire scene". I don't think this is true. Can the authors give some support for this claim?

The paper says that it wants the model to "effortlessly" estimate camera poses and estimate novel views. What does this mean? How can we distinguish effortless vs effortful models?

In talking about pose estimation vs. view prediction, the paper says "human cognition does not perceive these processes as isolated entities". I think it's quite clear that this is false. If we did not perceive these as separate tasks, how could we even talk about them so distinctly?

The paper says that their approach is "a model designed to align its understanding of 3D space with that of humans". Given that I question the paper's description of human cognition, I also question whether it's a good idea to design a model based on this shaky foundation.

The paper makes a major claim about novelty by saying "we introduce, for the first time, the concept of tokenizing the camera". It is not the first time. The authors can read, for example, "Input-level inductive biases for 3d reconstruction" (CVPR 2022). There are probably dozens of papers that convert camera information into tokens; checking the papers that cite the input-level biases paper should reveal many references. The paper goes on to say, "Specifically, we leverage Plucker coordinates to transform the camera into a camera map akin to an image". This has also been done before: check "Cameras as Rays: Pose Estimation via Ray Diffusion" (ICLR 2024). I can see that this is in fact cited in a later part of the paper, so perhaps there was some mixup in the writing, and the author of this section is not aware of the related work and what the actual contributions are. Nonetheless, this is a serious issue.

Overall, it is unfortunate for a work like this, which appears to be otherwise quite well polished, to make wild and unsupported statements about human cognition, and to claim novelty on techniques that have been published before (and even gained re-use across follow-up work). It is possible that this paper actually has novel parts and useful contributions, but the amount of false (or unsupported) and misleading material is overwhelming.

the Al puzzle -> the AI puzzle

**Questions:**

Please interpret some of my stated "weaknesses" as questions.

---

> ### Author Response · Authors · 2024-11-20
> **Response to reviewer VC2z**
>
> We sincerely thank you for your thoughtful review and valuable feedback on our manuscript, and we are grateful for the opportunity to clarify and improve our paper based on your comments. We have carefully considered your concerns and have addressed them point by point below:
>
> **1. Human Capability in Scene Reconstruction**
>
> *Response*: Thank you for bringing this to our attention. Our original text was: "*For humans, commencing from the observation of an image, they can effortlessly reconstruct the entire scene represented by the image in their minds*". Our intention was to convey that humans have a remarkable ability to infer spatial relationships and imagine unseen regions of a scene based on limited visual information and prior experiences [1,2].
>
> We acknowledge that the phrase "reconstruct the entire scene" may overstate human capability. We have revised the manuscript (Line 59) to more accurately reflect this by stating that humans can "infer spatial layouts and predict unseen aspects of the environment" from a single image.
>
>  **2. Use of the Term "Effortlessly" for Models**
>
> *Response*: By "effortlessly," we intended to express that our model performs both tasks seamlessly within a unified framework, without the need for separate models or additional task-specific components.
>
> To avoid ambiguity, we will replace "effortlessly" with "effectively" in the manuscript. This change more accurately reflects our goal of developing a model that jointly handles spatial localization and view prediction.
>
> **3. Perception of Pose Estimation and View Prediction**
>
> *Response*: Thank you for this insightful comment. Our assertion was based on the understanding that, while pose estimation and view prediction can be studied as distinct cognitive processes, humans often integrate these processes subconsciously during spatial reasoning.
> The absence of definitive research demonstrating the utilization of distinct brain regions for these two tasks by humans motivates our integrated approach. We have revised the manuscript to clarify that, although pose estimation and view prediction are distinct tasks, they are closely interconnected in human spatial cognition (Line 74). This integrated approach inspired us to develop a model that jointly addresses both tasks within a unified framework.
>
> **4. Designing Models Based on Human Cognition**
>
> *Response*: In fact, many AI models are indeed designed based on observations of human capabilities. For instance, the architecture and learning algorithms of neural networks are largely inspired by the interactions and adaptations of neurons in the brain. The Long Short-Term Memory (LSTM) structure draws inspiration from human memory mechanisms. Our objective is to develop a model inspired by human spatial reasoning abilities, where related tasks are typically processed jointly.
>
> We have revised the manuscript to modify any misunderstanding about human cognition, replacing them with our observations of human spatial reasoning. We will emphasize that our approach is inspired by (Line 76), rather than a direct replication of, human spatial reasoning abilities. This ensures that our model design is grounded in established cognitive principles without overextending the comparison.
>
> **5. Novelty Claim Regarding Tokenizing the Camera**
>
> *Response*: Our original text was: "*we introduce, for the first time, the concept of tokenizing the camera and incorporating it into the training of the auto-regressive model*". Our intention is to convey that, to the best of our knowledge, **we are the first to introduce the incorporation of camera as a new modality into the training and inference processes of the auto-regressive model that can serve as both input and output**. To achieve this, we trained a VQVAE that tokenizes camera into token sequences, drawing inspiration from the methodologies employed by language models in processing textual tokens. This differs from your understanding of "we are the first to tokenize the camera."
>
> And we acknowledge that prior works have explored tokenizing camera parameters or representing camera information for neural network inputs. We have revised the manuscript (Line 79) to accurately reflect the novelty of our approach, clearly positioning our contributions in relation to existing works.
>
> **6. Typographical Error**
>
> *Response*: Thank you for catching this error. We will correct the typo in the revised manuscript.
>
> Your feedback has been invaluable in helping us clarify our statements, correct inaccuracies, and better articulate the contributions of our work. We will incorporate the necessary revisions to enhance the quality and clarity of our paper. Again, we sincerely thank you for your constructive comments and for giving us the opportunity to refine our work.
>
> [1] Peelen et al. Predictive processing of scenes and objects. 2024.
>
> [2] Epstein et al. The cognitive map in humans: spatial navigation and beyond. 2017.

---

> ### Author Response · Authors · 2024-11-24
> **Official Comment by Authors**
>
> Dear Reviwer VC2z,
>
> I greatly appreciate the time and effort you have dedicated to evaluating our work and the constructive feedback you provided.
> Following your review, we have submitted an official comment and a revised manuscript addressing the concerns raised.
> We hope that our response adequately addresses your concerns.
> As the discussion phase draws to a close, we sincerely appreciate your feedback and would like to know if there are any additional issues we can address. We would be extremely grateful if you could consider raising the score of our paper. Thank you once again for reviewing our work!
>
> Best regards,
>
> GST Authors

---

> > ### Comment · Reviewer_VC2z · 2024-11-25
> >
> > Thanks. I appreciate most of this, but the response to "Designing Models Based on Human Cognition" is weird to me. It's responding to something I did not say, and ignoring what I said. I did not say it's bad to take inspiration from human cognition. I said that the paper's summary of scientific understanding on human cognition is shaky, and therefore designing based on this understanding is a bad idea.

---

> > > ### Author Response · Authors · 2024-11-25
> > > **Response to reviewer VC2z**
> > >
> > > Thank you for your valuable feedback.
> > > We believe that the presence of your concerns stems from a gap between the message we intend to convey and what was articulated in the original paper.
> > >
> > > We acknowledge that, in the original manuscript, some of our descriptions of human cognition were overstated, which may have led to ambiguity. Therefore, in the revised paper, we have been careful to remove related descriptions that might suggest a shaky scientific understanding of human cognition.
> > > In fact, our intention is not to provide a definition of human cognition, but rather to elucidate the motivation behind our tasks and methods.
> > > In the revised paper, we have refined our discussion to focus on our observations of human spatial reasoning (**Line 59, 74, 76**), rather than making broad claims about human cognition.
> > >
> > > Furthermore, we have explicitly clarified in the introduction section  (**highlighted in orange**)  that our observations regarding human spatial reasoning ability, rather than scientific understanding, and our model is inspired by these observations, which is the original intent of our work.
> > >
> > > We hope this revision satisfactorily addresses your concerns.
> > > Should you have additional queries or require further clarification, we would be happy to provide it. Thank you once again for your insightful comments!

---

> ### Author Response · Authors · 2024-11-29
> **Official Comment by Authors**
>
> Dear Reviwer VC2z,
>
> We sincerely appreciate the time and effort you have dedicated to evaluating our work. As we approach the end of the discussion phase, we would like to know if our responses have addressed your questions and concerns. We look forward to your feedback and would like to know if there are any other issues we can address. Thank you once again for reviewing our work!
>
> Best regards,
>
> GST Authors

---

### Author Response · Authors · 2024-11-23
**General Response**

Dear Reviewers and Area Chair,

We sincerely appreciate the insightful and detailed feedback provided by the reviewers. Your comments have been instrumental in refining our paper and further enhancing the clarity and depth of our research.

We sincerely appreciate the reviewers for their affirmations of our paper: The innovative approach of the **G**enerative **S**patial **T**ransformer (GST) and its dual capability in spatial localization and view prediction tasks have been recognized as novel and intriguing (**Reviewer AKff, 3EtM, xnP6**). Utilizing a single model to establish the joint distribution of camera pose and viewpoint images for novel view synthesis and camera pose estimation is a concise and effective approach (**Reviewer 3EtM, xnP6**), revealing the intrinsic connection between spatial awareness and visual prediction (**Reviewer AKff**). Moreover, leveraging VQVAE to tokenize camera views in a manner consistent with image tokenization is considered novel (**Reviewer xnP6**). Finally, the clear textual presentation and the exquisite visual illustrations in this article are widely acknowledged (**Reviewer VC2z, AKff, 3EtM, xnP6**).

Furthermore, in the revised manuscript, both in the main body and supplementary materials, we have addressed the concerns raised by the reviewers and made corresponding improvements, highlighting these changes **in orange font**.

We have made every effort to address the concerns raised during the review process comprehensively. We believe these revisions have significantly strengthened our paper, highlighting the novelty, effectiveness, and applicability of our approach more clearly. We thank the reviewers once again for their constructive criticisms and invaluable input, which were crucial for these enhancements.

---

### Meta-Review · Area_Chair_7EKT · 2024-12-19

**Metareview:**

This paper trains a joint model for novel-view synthesis and camera pose estimation. The cameras are parametrized via (tokenized) raymaps and novel-view images via VQ-VAE tokens.  Given a single input image, a common autoregressive transformer  predicts tokens for novel-view image patches and/or camera raymaps. The experiments demonstrate that this single network is better than/on par with prior approaches tacking a single task.

On the positive side, the approach of the paper is interesting both technically and practically — it is desirable to have a single model that can perform the two tasks of pose estimation and novel-view synthesis, and the technical contribution of using autoregressive models to jointly capture pose and novel views is a positive.

However, the current formulation is limited to a single input image, and cannot be trivially extended for multi-view (unposed) input. Moreover, based on results, the current method is worse than RayDiffusion when trained on similar data (perhaps highlighting the limitation of an autoregressive approach).

Overall, the central idea of the paper is neat, but there are also obvious limitations of the approach (single-view input) and concerns about the efficacy of the method. On the balance, the AC feels the technical contributions maybe interesting enough to justify acceptance and encourages the authors to improve the final text based on the reviewer feedback.

**Additional Comments On Reviewer Discussion:**

R-xnP6 raised some initial concerns about the view synthesis results and precision of camera poses, and these were well-addressed in the response. There was some discussion with R-VC2z about the positioning of the paper, and R-VC2z remained unconvinced. While the concerns raised are valid and a slight negative, perhaps the bigger concerns were pointed out in the discussion with R-3EtM (performance vs RayDiffusion, handling multi-view input). On the balance, these concerns are valid but perhaps the paper is technically interesting enough to accept despite these.

---

### Decision · Program_Chairs · 2025-01-22

Accept (Poster)